# Financial Distress Prediction of Cooperative Financial Institutions—Evidence for Taiwan Credit Unions

Chien-Min Kang [1,*] , Ming-Chieh Wang [2] and Lin Lin [3]

1    The Ph.D. Program in Strategy and Development of Emerging Industries, College of Management,
    National Chi-Nan University, 1 University Rd., Puli, Nantou 545, Taiwan
2    Department of International Business Studies, National Chi-Nan University, Nantou 545, Taiwan;
    mcwang@ncnu.edu.tw
3    Newhuadu Business School, Min-Jiang University, Fuzhou 350108, China; linlin.ncnu@gmail.com
*    Correspondence: arickkang@gmail.com

**Abstract:** In response to relatively little evidence on the determinants of the financial distress in
cooperative financial institutions (e.g., Credit Unions), this paper proposes a distress indicator of
Merton Distance to default (Merton DD), which was constructed with a z-score, possessed improved
predictive capability, but reducing equity volatility. This model possesses the advantages of both
hazard and modified Merton DD model, which could timely reflect market volatility and predict
when distress would occur. As a demonstration, we applied this model to forecast the financial
distress of credit unions in Taiwan. The results can provide more information to researchers.

**Keywords:** credit unions; financial distress; hazard; Merton DD





## 1. Introduction

Academic research on the determinants of corporate survival or failure stems from
Beaver's (1966, 1968a, 1968b) univariate model to assess the differences between surviving
and non-surviving firms. Subsequently, Altman (1968, 1993) mentioned the importance
of measuring leverage degrees of companies, proposing Altman's "Z-score" according to
which the leverage degree was the ratio of the market value of equity and book value of
liabilities.

Following this, Deakin (1972) and Edminster (1972) studied and developed multiple
discriminant analysis (MDA), whereas Ohlson (1980) utilized the logistic model to develop
a model of financial distress, using an "O-score" to construct default prediction. From
here, the Logit and Probit model was developed, which was a qualitative-response model.
Jones (1987) and Hillegeist et al. (2004) followed the same direction in later proposing a
model of corporate default probability, discussing certain variables of companies and the
degree of co-variance as time passes. Pille and Paradi (2002) developed data envelopment
analysis (DEA)-based models to detect weaknesses in credit unions in Ontario, Canada, so
that potential failures can be predicted. They conclude that a simple equity/asset ratio is
a good predictor of failure and is not improved upon by the more complex Z-score and
DEA-based models. The limited evidence on credit union failure suggests that young,
small, and poorly capitalized credit unions are most likely to fail (Kharadia and Collins
1981; Goddard et al. 2009).

Chen et al. (2007) show a directional distance function is used to construct adjusted
Malmquist–Luenberger productivity indexes, which simultaneously account for the im-
pacts of undesirable outputs, environmental variables, and statistical noise. Panel data
for 263 farmers' credit unions in Taiwan covering the 1998–2000 periods are employed to
illustrate the advantages of this method. This resulted in the proliferation of studies using
logit analysis and an improvement in financial distress predictability (Campbell et al. 2008;
Sun et al. 2014; Jones et al. 2015, 2017). Furthermore, Zmijewski (1984) employed probit

analysis and developed a three-variable distress prediction model, which was further tested by many researchers, including Wu et al. (2010) and Kleinert (2014).

Goddard et al. (2009) researched the determinants of liquidation or acquisition for U.S. credit unions during the 2001–2006 period and found that the risk of credit unions terminating operations has an inverse relationship with asset scale and profitability and a positive relationship with asset liquidity. With limited growth, credit unions are less likely to be the targets of acquisition, but more likely to be in financial distress. The researchers continued to add new variables to the distress prediction literature with a strong theoretical background. For instance, Tykvová and Borell (2012) employed a set of liquidity, profitability, and solvency ratios; moreover, Korol (2013) used a set of profitability, liquidity, and activity ratios with a strong theoretical background. In addition to the statistical-based techniques, several artificial intelligence modeling techniques, including support vector machines, genetic algorithms, decision trees, and neural networks, have been largely developed in recent years. Much of the financial distress literature has relied on quite simpler prediction accuracy methods, as they are a better predictor of financial distress (Jones et al. 2017); therefore, we also restricted our study to the statistical-based techniques.

In this study, we seek to fill this gap by examining the determinants of financial distress activity for the Taiwan credit union movement. Most previous studies find the factors leading to financial distress. The contribution of this paper is the incorporation of Merton Distance to default (Merton DD) variables into a model of the determinants of the probability of financial distress for a credit union. Bharath and Shumway (2008) found that the hazard of disappearance is inversely related to both asset size and profitability, and positively related to liquidity and modified Merton DD. It adopts the unique internal data of Taiwan's credit unions to carry out the model estimation. The data of financial and organizational variables were retrieved from the league of credit unions in Taiwan from January, 2001 to December, 2009. We aim to answer three questions in this paper: (1) Do financial distress prediction models have the ability to predict the financial distress of cooperative financial institutions with early warning signs of bankruptcy? (2) Which financial distress prediction model (Cox (1972), proportional hazard model; Merton (1974), Merton DD model; Bharath and Shumway (2008), naïve prediction indicator model, and logit model) can predict financial distress of cooperative financial institutions (credit unions of Taiwan) more accurately? and (3) What is the predictive ability of the models during the financial crisis?

The first issue is that the financial distress prediction models have the ability to predict the financial distress of cooperative financial institutions with early warning signs of bankruptcy. The early warning signs are profitability, asset-scale, liquidity, capitalization, operating expenses to total assets rate, and salaries to total income rate of cooperative financial institutions. The second issue is that we confirmed the accuracy and contribution of both the hazard and the modified Merton DD model, which could timely reflect market volatility and predict when distress would occur. Thirdly, a financial distress prediction model that can prevent credit unions from facing financial distress will combine the hazard model with a corrected Merton DD. This combined model can reflect market volatility information in real-time and predict the time point when distress occurs.

The rest of this paper is structured as follows. Section 2 provides a literature review of the development of financial distress prediction. Section 3 develops the theoretical background and model comparison of the study, and Section 4 presents the data and model specifications. In Section 5, the author discusses the empirical evidence and results. Finally, the last section gives the conclusions.

## 2. Literature Review

In this section, we provide a selective review of academic literature on the situation of the credit union movement and banking and the adoption and diffusion of the Merton DD model.

### 2.1. The Credit Union Movement and Banking

Credit unions are not-for-profit financial cooperatives. Each credit union is governed by its members, who elect from within the membership unpaid volunteer officers and directors. Voting is on a one-member-one-vote basis, regardless of the size of each member's financial stake (Goddard et al. 2009). At the end of 2020, there were 86,451 credit unions in 118 countries all over the world, with a membership of 375 million and total assets of $3208 billion. In recent years, the asset and membership of credit unions have grown, but the number has declined through consolidation. As credit unions have become larger and more sophisticated, there has been a gradual shift away from using volunteers for day-to-day operational needs and toward salaried employees. Credit unions serve a membership defined theoretically by a common bond (Goddard et al. 2002, 2008b). The common bond might restrict membership to members of a local community, employees of a particular firm, or individuals with some other organizational affiliation.

Growth in membership has also been accompanied by product diversification, particularly in the case of the larger credit unions (Goddard et al. 2008a). Many credit unions provide an array of retail financial services similar to those of banks and savings and loan associations. In addition, credit unions may also offer interest-bearing business checking accounts and commercial loans, agricultural loans, and venture capital loans. Credit unions may also deal in investment products such as bankers' acceptances, cash forward agreements, and reverse purchase transactions. These product offerings have further blurred the lines of demarcation between credit unions and mainstream financial services providers (Tokle and Tokle 2000; Feinberg 2001; Feinberg and Rahman 2001; Schmid 2006).

Estrella et al. (2000) find that in the case of the leverage ratio, the denominator is the total assets of the bank. This measure, which has a long history, assumes implicitly that the capital needs of a bank are directly proportional to its level of assets. For some broad classes of banks, this may not be a bad assumption; however, if we take the example of two banks, only one of which has substantial and risky off-balance sheet activities, the use of the leverage ratio may produce misleading relative results. An unadjusted capital to gross revenue measure, suggested by Shepheard-Walwyn and Litterman (1998), performs reasonably well in predicting bank failure. The studies have examined the role of bank-specific, regulatory and regional economic conditions as determinants of bank failure (Wheelock and Wilson 1995, 2000; Cole and Gunther 1995, 1998; Kolari et al. 2002; Nuxoll et al. 2003; King et al. 2006; and Lanine and Vander Vennet 2006).

### 2.2. Merton DD Model

The shift to a market-based model was made by Black and Scholes (1973), Merton (1974), Fisher et al. (1989), and Leland (1994), all of whom indicated that the process of asset formation follows the rules of geometric Brownian motion. In the early development of this model, distance-to-default was used to measure the conditional default probability of companies and was calculated by the standard deviation of the annual asset growth rate; it showed that the asset value of a company was greater than its liabilities.

Stein (2000), Sobehart and Stein (2000), and Sobehart and Keenan (1999, 2002a, 2002b) doubted that Merton DD models could effectively make predictions; Kealhofer and Kurbat (2001) even doubted that the Merton DD-like model developed by KMV corporation could show information such as traditional credit ratings and accounting statements. Crosbie and Bohn (2002) and Kealhofer (2003) pointed out that Moody's KMV empirically conducted the prediction of default probability on listed companies by adopting the covariance relationship between the market value of corporate equity and the information of liabilities. Duffie and Lando (2001) stated that distance-to-default may not be accurately measured by the KMV, and that the degree of default probability depends on the measurement of the distance-to-default, among other possible variables. The determinants that affect corporate conditional default probability are those that influence the financial sturdiness of companies, such as industry categories, the distribution of product categories, and

overall environmental variables, all of which influence the profits and leverage degree of companies.

Duffie et al. (2007) indicated that the Merton DD probability prediction model possesses significant predictive capability under a consecutive time series of default probability. Campbell et al. (2008) also pointed out that when the indicator π considers other variables, which are related to bankruptcy in hazard models, its predictive capability on bankruptcy becomes relatively weaker.

In contrast to previous models, recent default probability prediction models emphasize duration. Lane et al. (1986) initiated this development by proposing time-independent covariates aiming at analyzing the default probability of banks. They favored the Cox proportional-hazard model, because the survival model (SM) was not as prevalent as mainstream models such as MDA and the logit model, and the prediction accuracy of SM was just about the same as out-of-sample forecasts performed by MDA; SM also had lower Type I errors. When Crapp and Stevenson (1987) used the Cox model to analyze credit unions in Australia, they discovered similar findings to those stated above. Lee and Urrutia (1996) proposed the duration model using the Weibull distribution of default times to predict bankruptcy probability in insurance companies and, at the same time, compared the difference between duration and logit models. They discovered that the duration model was more capable of detecting significant variables than the logit model. Other studies in duration analysis by Shumway (2001), Chava and Jarrow (2004), and Hillegeist et al. (2004) were also used to predict bankruptcy.

Shumway (2001) used the discrete duration model, which has time-dependent covariates. This was the first time in empirical examination that standard errors were adjusted to run multi-period logit model analysis. Likewise, Shumway (2001) was the first to discover that the current survival model (SM) theory outperformed the traditional mainstream MDA and logit models. Empirically, in out-of-sample forecasts, Shumway SM was obviously better than the MDA and logit models, which also considered Type I errors; however, their errors were below 10% and also below the real-world situation. His model was further tested by a number of researchers, including Campbell et al. (2008) and Bonfim (2009). Later on, researchers including Chava and Jarrow (2004) and Agarwal and Taffler (2008) articulated that market-based variables reflecting both internal and external information increase the overall predictability of distress prediction models. Further, Trujillo-Ponce et al. (2014) suggested that a combined model with both accounting and market-based variables is the best option, as both types of information are important for distress prediction.

In predicting a one-year default, Hillegeist et al. (2004) adopted the same method. They utilized the Black–Scholes–Merton model to calculate and estimate theoretical default probability and discovered that the distance-to-default was insufficient to explain the variation of corporate default probability. Bharath and Shumway (2004) and Campbell et al. (2008) also discovered that when taking the degree and variances of corporate leverage in the market into account, a measure of the distance-to-default could not offer sufficient default information. Bharath and Shumway (2008) found out that the naïve prediction indicator was better than hazard models, and that it had better performance than both the Merton DD model and reduced-form model in terms of out-of-sample forecasts.

## 3. Theoretical Development and Model Comparison

There are some differences among the Merton DD, logit, and hazard models. First, most previous literature used the Merton and KMV models to investigate financial distress, but the result showed poor predictive capability (Stein 2000; Sobehart and Stein 2000; Duffie and Lando 2001; Kealhofer and Kurbat 2001; Sobehart and Keenan 1999, 2002a, 2002b; Bharath and Shumway 2004). To determine when a company will have financial distress, the Merton model uses the standard of total assets below total liability, and the KMV model uses the standard of total assets below the sum of current liability and half long-term liability; however, neither models' definitions of default take into account default events such as a bounced check, a run on banks, relief and help, or suspension due to financial

distress. Those events are classified as indicators of insufficient liquidity during operation, rather than bankruptcy, which has a narrower definition. In addition, the financial crises of many companies include difficulties in operation such as too much inventory, too many payable accounts, and difficulties in capital management. Those difficulties reflect short-term categories on financial reports, including the change in current assets and current liabilities.

The biggest difference between the Merton DD model and other models is that the former has a daily hazard measurement period, which allows it to follow the market condition at any time, and precisely controls the credit hazard change of asset positions. In addition, the Merton DD model does not use credit rating data but instead uses the financial and stock data of individual companies to calculate the company's expected default frequency. This improves the accuracy of the model forecast, and the data used as variables are easier to find. The Merton DD model uses option valuation theory to evaluate a company's credit hazard and it assumes that a change in the company's asset value is displayed in normal distribution during the process; however, the empirical result shows that the loss distribution of a company's default events is often non-normal. The Merton DD model also assumes the occurrence of default events when the liabilities are due, which is irrational; these two situations violate the assumption of the Merton DD model. When a company is facing closure, the asset value might tighten rapidly due to quick disposal, or the company might try its best to adjust the proportion of liability and assets during the period of financial distress. Thus, considering the volatility of liability is the more realistic approach (Bharath and Shumway 2008).

Compared with the traditional MDA model, the logit model does not assume a variable distribution method. Thus, the empirical research does not need to concern itself with whether the variable distribution method obeys the normal assumption or not. In addition, since there is no problem with normal assumptions, the logit model can use dummy variables to perform the regression research. The model has more options for the set of variables and is able to find the most suitable explanatory variable. The logit model originally evolved from the linear probability model, which evaluates the probability of events; however, the biggest defect of the linear probability model is that the probability of events may be larger than one or smaller than zero, because the model uses the linear method to perform the regression evaluation of the event probability. Since the result of the event fits in a straight line, it may exceed the restriction scope of probability, which is an irrational probability result of default estimation. The logit model uses the linear regression method to calculate the event probability and then transfers the result with the logistic distribution, so the final probability obeys the form of probability and remains consistent with the original regression result. For this reason, empirical research favors the logit model over the linear probability model.

The logit model improves the defect of the MDA model, which pertains to the process and distribution of nominal variables; however, the logit model still cannot provide the default probability (or survival rate) of the research target at different times in the future. Using a survival analysis (widely used in the medical and actuarial fields) as the foundation for prediction establishes a complete company hazard data record and confirms the implementation applicability of this research. The advantage of the hazard model (hazard and survival analysis) is the ability to forecast the time point close to default. Although the logit model can predict the default probability for a period in the future, it is unable to predict the time point close to default and it lists the company as dangerous before it is exposed to the default hazard, which easily results in the loss of important customers. By contrast, the survival analysis forecasts the time point close to default and the period of time when a company will be in a high hazard for default. Previous literature has shown that the Cox model has fewer type I errors, which reduces the credit error loss of financial institutions. Since more significant variables can provide more accurate information for prediction, the usage of the survival model is suggested; however, the different models have evolved according to different characteristics of research data, and together they provide a wide

usage scope with smaller assumption restrictions. The Merton DD model concerns the real-time market data; the logit model is practical and efficient; the hazard model predicts the time point of default according to variations over time. Thus, each model has its scope and restriction of usage, which depends on the purpose of research and the form of the data.

As mentioned in the introduction, this research tests certain assumptions pertaining to financial distress prediction (FDP) models. More specifically, it examines the statistical and economic significance of the Merton DD model default probability ($\pi\_merton$), a simplified alternative Shumway ($\pi\_shumway$) probability, and a revised alternative $\pi\_shumway$ probability (Bharath and Shumway 2008). First, the various models are explained, beginning with the Merton model. The theoretical development of the Merton DD model is described in detail in Appendix A.

Credit unions do not collect capital by issuing shares to the public; the capital is only the combined deposits of internal members. Thus, when defining the volatility of shareholder equity, one cannot collect the data according to the estimated condition of the original Merton model. Calculating the volatility requires replacing the stock price with net income per month divided by deposited shares. To determine the effects of global financial distress on equity volatility, this research calculates the equity volatility on the basis of monthly data before 2007 and after 2008.

The above method calculates the $\pi\_shumway$ according to the value at which the term structure and equity volatility are set and develops four simplified probabilities. Those simplified probabilities define $\pi$ (term structure volatility, equity volatility), such that $\pi\_\alpha\_up$ (0.5,0.25), $\pi\_\alpha\_down$ (0.005,0.25), $\pi\_\beta\_up$ (0.05,2.5), and $\pi\_\beta\_down$ (0.05,0.025) (Bharath and Shumway 2008). Through the estimation of simplified probability mentioned above, a better predictive description of credit union financial distress will be found. The suitable settings for the credit unions in Taiwan can be found by examining the data characteristics of credit unions and the adjustment of term structure volatility and equity volatility.

## 4. Data and Model Specification

### 4.1. Data and Sample Selection

We describe the data that are used below to estimate hazard functions for financial distress[1] of Taiwan credit unions. We also discuss the selection of covariates for the hazard functions. The credit union balance sheet and income statement unique internal data are compiled from the annual reports published by the League of Credit Unions in Taiwan (CULROC). Data are available for the period January 2001 to December 2009, providing time-series observations on each credit union. At the end of 2009, there were 336 credit unions in Taiwan, with around 201,486 members and about 23.5 billion NT dollars in total assets.

The panel data of all credit unions every month and every year were retrieved, and different default estimation models were used to predict the possibility of financial distress in these credit unions. This research attempts to see whether credit unions in Taiwan have a different financial distress prediction system from those shown in the literature. Generally, companies are evaluated by credit rating institutions, but credit unions do not have a similar professional institution to evaluate them. Their financial operation is neither public nor transparent, so they are usually unable to reverse the situation when financial distress strikes.

Table 1 provides the definitions of covariates. The covariates of the hazard functions are age, asset size, asset quality, financial risk, managerial efficiency, growth, profitability, macro factor, asset volatility, and expected default frequency (EDF). In addition, we include the region and common bond characteristics of each credit union, distinguishing between eastern and western credit unions, and between single and multiple common bond credit unions.

**Table 1.** Covariates definition.

| Category | Covariate | Definition |
|---|---|---|
| region_type | EAST | region(west-0,east-1) (dummy identifying the eastern credit unions) |
| common_bond | MULT | single-0,multiple-1(dummy identifying multiple common bond credit unions) |
| age | LAGE | natural logarithm of (current year—year of formation) |
| size | LASSET | natural logarithm of total assets |
| asset_quality | CAP_ADE | net worth/total assets (capital to assets ratio) |
| asset_quality | LOAN_RATIO | total loans/total assets |
| asset_quality | LOAN_COV | overdue_loan_Coverage (overdue_reserves/overdue_loans) |
| financial risk | LIQ | current_assets/total_assets |
| financial risk | RES_CAP | reserves/total_loans |
| managerial_efficiency | OPE_EFF | operating_expenses/total_assets |
| managerial_efficiency | INC_CAP | total_income/total_assets |
| managerial_efficiency | LAB_COST | salaries/total_income |
| growth | MEM_GRO | membership growth (membership change/members at start) |
| growth | LOAN_GRO | loan growth (loan change/loans at start) |
| growth | SHARE_GRO | share growth (share change/shares at start) |
| profitability | STO_RET | stock return |
| profitability | ROA | net income/total assets (return on assets) |
| macro_factor* | M1b | money supply growth rate |
| macro_factor* | GDP | annual average rate of GDP |
| macro_factor* | RATE | average interest rate of deposits(local banks) |
| macro_factor* | RATE_SPR | spreads from deposit and loan (local banks) |
| expected default frequency | $\pi$_merton | Merton DD expected default frequency |
| expected default frequency | $\pi$_shumway | Merton DD Shumway expected default frequency |
| asset volatility | $\sigma v$_merton | Merton DD standard deviation |
| asset volatility | $\sigma v$_shumway | Merton DD Shumway standard deviation |

Note: 1. The covariates of the hazard functions are age, asset size, asset quality, financial risk, managerial efficiency, growth, profitability, macro factor, asset volatility, and expected default frequency (EDF). In addition, we include the region and common bond characteristics of each credit union, distinguishing between eastern and western credit unions, and between single and multiple common bond credit unions. 2. The covariates selection and definition followed by Crapp and Stevenson (1987) and Goddard et al. (2009). 3. Macro_factor from Accounting and Statistics in Taiwan.

Financial distress, defined by the *Taiwan Economic Journal* (TEJ), can be divided into two categories: financial distress and quasi-financial distress. Thus, CULROC provides the criteria of negative net worth at the end year as financial distress for credit unions; this study uses this financial distress criterion (Lin 2009; Chen and Du 2009; Lin et al. 2011; Cheng et al. 2019; Chen et al. 2020).

In 2001, CULROC reported data for 353 credit unions. We eliminated from the sample credit unions as follows: First, credit unions that reported an extreme value for any variable; second, credit unions at the year of formation; third, credit unions that had missing data. Trimming the sample resulted in financial distress of 10 credit unions (3.3% of the total reported for 2001). The final sample comprises 297 credit unions that did not have financial distress in 2001, of which 261 survived until 2009 and 53 had financial distress prior to 2009.

### 4.2. Estimation Method

Wheelock and Wilson (2000) pointed out that the estimation of hazard functions is commonly used when researching the factors of default or merger/acquisition in the American banking industry. The estimation of hazard functions for the financial distress of Taiwan credit unions is based on the method used by Goddard et al. (2009) to model the determinants of failure and acquisition for US credit unions. The empirical model for the hazard of financial distress is based on the Cox (1972) proportional hazard model with time-varying covariates.

Hazard or survival function techniques are often used in the data of lifetime distribution and are applied to dynamic statistical tools in order to analyze the probability of a certain event occurring at a given time. Thus, using these analysis techniques in financial

distress prediction implies a departure from traditional models. While other analysis techniques focus on the error cost of classification, the survival functions technique focuses on lifetime distributions. These distributions represent the distribution of random variables of a non-negative number and the lifetime of the individual sample in the parent body. The survival functions S(t) represent the probability of a company remaining in survival at time t; the hazard functions h(t) represent the probability of a company synchronously closing down at time t.

Previous research has been more interested in comparing the survival probability of two or more groups. For instance, in the default research on company bankruptcy, the different operating methods between ordinary companies and bankrupted companies are compared; however, an unbiased comparison is not always possible, since, in most cases, the individuals within a group have unique features that may affect the research result. For example, variables such as ROA and the equity pledge ratio of shareholders can be seen as the covariates (explanatory variables) of survival time. Comparisons of survival time or the probability of default bankruptcy will have fewer biases after the adjustment of these potential explanatory variables.

In the survival analysis, the most important regression analysis most often used is the Cox proportional hazards model, which is simply referred to as the Cox model:

$$\lambda\left\{t\,\middle|\,\overline{X}(t)\right\} = \lambda_0(t)\exp\{\beta X(t)\} \tag{1}$$

$$\overline{X}(t) = \{X(s) : 0 \leq s \leq t\} \tag{2}$$

All covariate history of the research target is before time *t*; $\beta$ is relative to the regression coefficient of covariate *X(t)*; $\lambda_0(t)$ is the unspecified baseline hazard function. In this model, the baseline hazard function can be explained as an individual's risk function whose covariate values all appear to be 0 at the time of origin and remain the same over time. In other words, covariate has no effect on the bankruptcy and default risk. The covariate *X(t)* and time *t* are interdependent. The regression coefficient $\beta$ can explain the increase in the logarithm risk function of two individuals at time *t*, and the integral of the risk function provides the survival function of an individual number *i*.

$$S_i\left(t\,\middle|\,\overline{X}(t)\right) = \exp\left[-\int_0^t \lambda_0(u)e^{\beta X_i(u)}du\right] \tag{3}$$

This survival function does not only connect with the unspecified baseline hazard function $\lambda_0$, but also relates to the time interdependent covariant value in the time interval $[0, t]$.

### 4.3. Choice of Hazard Function Covariates

When discussing which covariates of hazard functions to select in order to forecast financial distress in credit unions, this study explains the sample summary statistics for each of the covariates. The summary statistics are reported in Tables 2–5. Table 2 reports sample means and standard deviations for the time-varying covariates of the hazard function model. To calculate and illustrate these summary statistics, the annual observations on each sample credit union from the period 2001 to 2009 are pooled. Accordingly, each sample credit union contributes up to nine observations to the summary statistics. The more significant result is that the distribution of EDF calculated by the Merton DD model ($\pi$_merton) is similar to the distribution of simplified linear combination probability ($\pi$_shumway). The estimated mean of $\pi$_merton was collected from data of every credit union from 2001 to 2009 into pooled panel data. The estimated mean of $\pi$_merton collected from 2001 to 2009 is 8.835%, which is lower than 10.95%, the mean collected from 1980 to 2003 and calculated by Bharath and Shumway (2008); however, it is higher than the mean calculated from 1971 to 1999, which is 4.21%, according to Vassalou and Xing (2004). The correlation coefficient

of the EDF between the Merton DD model and the simplified model is 41.8%, which is statistically significant if under 1%; the distributions of asset volatility in the Merton DD and simplified models are similar as well, with the correlation coefficient reaching 43.9%.

**Table 2.** Summary statistics: Time-varying covariates. Means and standard deviations from CULROC.

| Variable | Mean | Std. Dev. | Minimum | Maximum |
|---|---|---|---|---|
| LAGE | 3.276 | 0.539 | 0.000 | 3.870 |
| LASSET | 3.705 | 0.968 | 0.400 | 7.290 |
| CAP_ADE (%) | 8.573 | 3.587 | 0.012 | 48.311 |
| LOAN_RATIO (%) | 52.910 | 18.316 | 5.583 | 95.978 |
| LOAN_COV (%) | 5.967 | 22.140 | 0.000 | 120.079 |
| LIQ (%) | 29.999 | 17.884 | 0.000 | 99.267 |
| RES_CAP (%) | 17.707 | 13.123 | 0.014 | 268.685 |
| OPE_EFF (%) | 2.778 | 1.496 | 0.020 | 30.616 |
| INC_CAP (%) | 5.214 | 2.765 | 0.045 | 75.574 |
| LAB_COST (%) | 14.690 | 8.504 | 0.000 | 80.517 |
| MEM_GRO (%) | 1.340 | 7.095 | −100.000 | 90.160 |
| LOAN_GRO (%) | −0.726 | 27.905 | −87.061 | 1062.873 |
| SHARE_GRO (%) | 2.462 | 8.711 | −57.939 | 132.582 |
| STO_RET (%) | 2.122 | 1.235 | 0.000 | 9.120 |
| ROA (%) | 2.481 | 2.341 | −23.058 | 49.318 |
| M1b(%) | 8.115 | 6.944 | −2.938 | 18.977 |
| GDP(%) | 2.392 | 3.136 | −2.492 | 6.388 |
| RATE(%) | 2.164 | 1.219 | 1.110 | 4.620 |
| RATE_SPR(%) | 2.479 | 0.473 | 1.760 | 3.150 |
| $\pi$_merton (%) | 8.835 | 19.229 | 0.000 | 100.000 |
| $\pi$_shumway (%) | 16.709 | 18.625 | 0.000 | 100.000 |
| $\sigma v$_merton (%) | 34.183 | 275.567 | 1.117 | 4897.295 |
| $\sigma v$_shumway (%) | 29.126 | 107.200 | 3.974 | 1959.254 |

Source of data: Credit unions Reports as filed with the CULROC. 1. This table reports summary statistics for all the variables used in the Merton DD model, the hazard models, and logit models. The covariates definition follows Table 1. 2. This table reports sample means and standard deviations for the time-varying covariates of the hazard function model. To calculate and illustrate these summary statistics, the annual observations on each sample credit union from the period 2001 to 2009 are pooled. Accordingly, each sample credit union contributes up to 9 observations to the summary statistics. 3. $\pi$_merton is the expected default frequency in percent and is given by formulas (A7) in the Appendix A. $\pi$_shumway is calculated according to formulas (A13) in the Appendix A. $\sigma v$_merton is annual asset volatility.$\sigma v$_shumway is calculated by formulas (A10) in the Appendix A. Our sample spans 2001 through 2009 and contains 33,516 monthly data of credit unions.

**Table 3.** Summary Statistics: Non-time-varying covariates from CULROC.

| | All Sample Credit Unions | Distress Credit Unions |
|---|---|---|
| Distribution by region_type | | |
| West | 0.813 | 0.817 |
| East | 0.187 | 0.183 |
| Distribution by common bond | | |
| Single | 0.164 | 0.172 |
| Multiple | 0.836 | 0.828 |
| Distribution by year of formation | | |
| 1961–1970 | 0.088 | 0.072 |
| 1971–1980 | 0.499 | 0.453 |
| 1981–1990 | 0.212 | 0.183 |
| 1991–2000 | 0.144 | 0.283 |
| 2001–2009 | 0.057 | 0.009 |

Source of data: Credit unions Reports as filed with the CULROC. 1. This table reports summary statistics for the non-time-varying covariates. These statistics illustrate the distributions by year of formation (age), region (region type), and relationship (common bond) for all the credit unions with or without financial distress.

**Table 4.** Mean values of time-varying covariates by observation: All sample credit unions from CULROC.

|  | 2001 | 2002 | 2003 | 2004 | 2005 | 2006 | 2007 | 2008 | 2009 |
|---|---|---|---|---|---|---|---|---|---|
| number | 307 | 308 | 308 | 310 | 310 | 311 | 313 | 314 | 314 |
| LASSET | 3.6300 | 3.6500 | 3.6700 | 3.7000 | 3.7200 | 3.7300 | 3.7400 | 3.7400 | 3.7500 |
| CAP_ADE | 0.0708 | 0.0716 | 0.0776 | 0.0815 | 0.0848 | 0.0887 | 0.0923 | 0.1005 | 0.1030 |
| LOAN_RATIO | 0.6403 | 0.6045 | 0.5591 | 0.5170 | 0.4978 | 0.5020 | 0.4976 | 0.4888 | 0.4573 |
| LOAN_COV | 5.9400 | 7.5700 | 7.7000 | 4.9500 | 6.9700 | 5.5800 | 4.9500 | 5.1200 | 4.9500 |
| LIQ | 0.2367 | 0.2553 | 0.2521 | 0.2661 | 0.2822 | 0.3133 | 0.3302 | 0.3540 | 0.4062 |
| RES_CAP | 0.1143 | 0.1209 | 0.1420 | 0.1669 | 0.1897 | 0.1940 | 0.2019 | 0.2216 | 0.2488 |
| OPE_EFF | 0.0292 | 0.0310 | 0.0275 | 0.0267 | 0.0279 | 0.0265 | 0.0262 | 0.0277 | 0.0274 |
| INC_CAP | 0.0732 | 0.0678 | 0.0538 | 0.0485 | 0.0485 | 0.0469 | 0.0455 | 0.0458 | 0.0396 |
| LAB_COST | 0.1123 | 0.1224 | 0.1467 | 0.1546 | 0.1518 | 0.1579 | 0.1533 | 0.1499 | 0.1725 |
| MEM_GRO | 0.0080 | 0.0103 | 0.0261 | 0.0249 | 0.0205 | 0.0140 | 0.0071 | 0.0048 | 0.0050 |
| LOAN_GRO | −0.0095 | −0.0193 | −0.0412 | −0.0434 | 0.0106 | 0.0671 | 0.0157 | 0.0044 | −0.0512 |
| SHARE_GRO | 0.0277 | 0.0288 | 0.0358 | 0.0420 | 0.0393 | 0.0162 | 0.0115 | 0.0079 | 0.0131 |
| STO_RET | 0.0388 | 0.0323 | 0.0238 | 0.0192 | 0.0181 | 0.0169 | 0.0170 | 0.0151 | 0.0103 |
| ROA | 0.0459 | 0.0376 | 0.0264 | 0.0221 | 0.0208 | 0.0210 | 0.0199 | 0.0181 | 0.0120 |
| M1b | 0.1058 | −0.0102 | 0.1701 | 0.1183 | 0.1898 | 0.0710 | 0.0530 | 0.0644 | −0.0294 |
| GDP | −0.0249 | 0.0487 | 0.0267 | 0.0638 | 0.0324 | 0.0432 | 0.0541 | −0.0135 | −0.0147 |
| RATE | 0.0462 | 0.0409 | 0.0238 | 0.0147 | 0.0111 | 0.0122 | 0.0140 | 0.0153 | 0.0171 |
| RATE_SPR | 0.0299 | 0.0290 | 0.0315 | 0.0263 | 0.0255 | 0.0245 | 0.0209 | 0.0183 | −0.0176 |
| $\pi$_merton | 0.0888 | 0.0823 | 0.0819 | 0.0817 | 0.0835 | 0.0843 | 0.0831 | 0.1042 | 0.1046 |
| $\pi$_shumway | 0.1391 | 0.1394 | 0.1574 | 0.1668 | 0.1718 | 0.1724 | 0.1689 | 0.1919 | 0.1950 |
| $\sigma$v_merton | 0.3395 | 0.3386 | 0.3386 | 0.3377 | 0.3385 | 0.3378 | 0.3359 | 0.3543 | 0.3543 |
| $\sigma$v_shumway | 0.2714 | 0.2750 | 0.2828 | 0.2861 | 0.2895 | 0.2929 | 0.2926 | 0.3151 | 0.3146 |
| $\pi$_$\alpha$_up | 0.4230 | 0.4288 | 0.4405 | 0.4464 | 0.4490 | 0.4502 | 0.4459 | 0.4568 | 0.4589 |
| $\pi$_$\alpha$_down | 0.1150 | 0.1136 | 0.1293 | 0.1372 | 0.1418 | 0.1426 | 0.1400 | 0.1631 | 0.1659 |
| $\pi$_$\beta$_up | 0.4614 | 0.4671 | 0.4915 | 0.5040 | 0.5102 | 0.5119 | 0.5040 | 0.5469 | 0.5471 |
| $\pi$_$\beta$_down | 0.0653 | 0.0676 | 0.0789 | 0.0850 | 0.0892 | 0.0876 | 0.0862 | 0.1017 | 0.1053 |
| Distress_rate | 0.0325 | 0.0357 | 0.0747 | 0.1226 | 0.1742 | 0.1576 | 0.1438 | 0.1592 | 0.1688 |

Note: 1. This table indicates the sample means of time-varying covariates collected from 2001 to 2009, including the data of all the credit unions. An increase in the mean of financial variables from 2001 to 2009 is indicated for the capital adequacy ratio (CAP_ADE), liquidity ratio (LIQ), the reserves capacity for loan ratio (RES_CAP), and labor cost ratio (LAB_COST), while a decrease is indicated for the loan ratio (LOAN_RATIO), income capacity rate (INC_CAP), stock return ratio (STO_RET) and ROA. 2. It takes the $\pi$_shumway according to the setting value of term structure and equity volatility and develops four simplified probabilities. Those simplified probabilities define $\pi$(term structure volatility, equity volatility factor), such as $\pi$_$\alpha$_up (0.5,0.25), $\pi$_$\alpha$_down (0.005,0.25), $\pi$_$\beta$_up (0.05,2.5), $\pi$_$\beta$_down (0.05,0.025).

Table 3 reports summary statistics for the non-time-varying covariates. These statistics illustrate the distributions by year of formation (age), region (region type), and relationship (common bond) for all the credit unions with or without financial distress. Table 4 indicates the sample means of time-varying covariates collected from 2001 to 2009, including the data of all the credit unions. An increase in the mean of financial variables from 2001 to 2009 is indicated for the capital adequacy ratio (CAP_ADE), liquidity ratio (LIQ), the reserves capacity for loan ratio (RES_CAP), and labor cost ratio (LAB_COST), while a decrease is indicated for the loan ratio (LOAN_RATIO), income capacity rate (INC_CAP), stock return ratio (STO_RET), and ROA. Table 5 shows that normally operated credit unions have a better liquidity ratio (LIQ), labor cost ratio (LAB_COST), coverage rate of overdue_loans (LOAN_COV), operating expense ratio (OPE_EFF), income capacity rate (INC_CAP), and ROA.

The $\pi$_merton of EDF in Table 4 shows a percentage of around 8% from 2001 to 2007, and it increased to 10% after 2008; the distribution of $\pi$_shumway had been shifting between 14% and 17% before 2007, while in 2008, it went up to 19%. Based on the adjustment of the volatility of term structure and equity, the market's equity volatility formed 4 EDF values, and in Table 4, only the distribution of $\pi$_$\beta$_down value is close to the distress rate; $\sigma$v_merton has little change and bounces only between 34% and 35%. The distribution of the $\sigma$v_shumway value before 2007 was between 27% and 29%, while it climbed up to

about 31.5% after 2008. Taking distress rate as the foundation, it is apparent that π_merton is slightly underestimated, and π_shumway is overestimated and still has space to revise down its value. In Table 5, each EDF and volatility score of financially distressed credit unions is clearly higher than that of non-distressed credit unions.

**Table 5.** Mean values of time-varying covariates by observation: Sample credit unions that are distressed and are not distressed during the subsequent one-year period from CULROC.

| | 2001 | 2002 | 2003 | 2004 | 2005 | 2006 | 2007 | 2008 | 2009 |
|---|---|---|---|---|---|---|---|---|---|
| No distress num | 297 | 297 | 285 | 272 | 256 | 262 | 268 | 264 | 261 |
| LASSET | 3.6400 | 3.6800 | 3.7300 | 3.7600 | 3.8000 | 3.8200 | 3.8100 | 3.8200 | 3.8600 |
| CAP_ADE | 0.0707 | 0.0714 | 0.0771 | 0.0808 | 0.0835 | 0.0872 | 0.0905 | 0.0981 | 0.0991 |
| LOAN_RATIO | 0.6399 | 0.6028 | 0.5647 | 0.5337 | 0.5147 | 0.5152 | 0.5045 | 0.4985 | 0.4567 |
| LOAN_COV | 5.7900 | 7.5100 | 8.1800 | 4.8200 | 7.4900 | 6.1200 | 5.6600 | 5.5500 | 5.5000 |
| LIQ | 0.2418 | 0.2616 | 0.2542 | 0.2633 | 0.2778 | 0.3098 | 0.3310 | 0.3546 | 0.4158 |
| RES_CAP | 0.1127 | 0.1212 | 0.1379 | 0.1560 | 0.1699 | 0.1773 | 0.1921 | 0.2080 | 0.2340 |
| OPE_EFF | 0.0295 | 0.0313 | 0.0275 | 0.0266 | 0.0273 | 0.0265 | 0.0257 | 0.0272 | 0.0257 |
| INC_CAP | 0.0744 | 0.0693 | 0.0554 | 0.0506 | 0.0507 | 0.0493 | 0.0472 | 0.0478 | 0.0411 |
| LAB_COST | 0.1089 | 0.1168 | 0.1367 | 0.1414 | 0.1327 | 0.1421 | 0.1383 | 0.1327 | 0.1505 |
| MEM_GRO | 0.0078 | 0.0117 | 0.0289 | 0.0283 | 0.0220 | 0.0164 | 0.0094 | 0.007 | 0.011 |
| LOAN_GRO | −0.0111 | −0.0166 | −0.0433 | −0.0412 | 0.0198 | 0.0164 | 0.0135 | −0.0043 | −0.0411 |
| SHARE_GRO | 0.0274 | 0.0317 | 0.0380 | 0.0406 | 0.0435 | 0.0211 | 0.0121 | 0.0107 | 0.0195 |
| STO_RET | 0.0395 | 0.0328 | 0.0250 | 0.0208 | 0.0198 | 0.0189 | 0.0186 | 0.0167 | 0.0114 |
| ROA | 0.0470 | 0.0388 | 0.0281 | 0.0244 | 0.0245 | 0.0238 | 0.0222 | 0.0206 | 0.0151 |
| M1b | 0.1058 | −0.0102 | 0.1701 | 0.1183 | 0.1898 | 0.0710 | 0.0530 | 0.0644 | −0.0294 |
| GDP | −0.0249 | 0.0487 | 0.0267 | 0.0638 | 0.0324 | 0.0432 | 0.0541 | −0.0135 | −0.0147 |
| RATE | 0.0462 | 0.0409 | 0.0238 | 0.0147 | 0.0111 | 0.0122 | 0.0140 | 0.0153 | 0.0171 |
| RATE_SPR | 0.0299 | 0.0290 | 0.0315 | 0.0263 | 0.0255 | 0.0245 | 0.0209 | 0.0183 | −0.0176 |
| π_merton | 0.0850 | 0.0812 | 0.0759 | 0.0685 | 0.0740 | 0.0705 | 0.0702 | 0.0967 | 0.0988 |
| π_shumway | 0.1352 | 0.1375 | 0.1473 | 0.1512 | 0.1529 | 0.1562 | 0.1526 | 0.1783 | 0.1874 |
| σv_merton | 0.3370 | 0.3430 | 0.3237 | 0.3149 | 0.3376 | 0.3142 | 0.1438 | 0.3554 | 0.3568 |
| σv_shumway | 0.2686 | 0.2754 | 0.2724 | 0.2709 | 0.2771 | 0.2749 | 0.2062 | 0.3023 | 0.3047 |
| π_α_up | 0.4223 | 0.4293 | 0.4340 | 0.4379 | 0.4384 | 0.4439 | 0.4362 | 0.4523 | 0.4562 |
| π_α_down | 0.1111 | 0.1119 | 0.1200 | 0.1220 | 0.1249 | 0.1271 | 0.1248 | 0.1503 | 0.1589 |
| π_β_up | 0.4563 | 0.4639 | 0.4758 | 0.4805 | 0.4793 | 0.4882 | 0.4798 | 0.5260 | 0.5354 |
| π_β_down | 0.0624 | 0.0664 | 0.0723 | 0.0742 | 0.0784 | 0.0763 | 0.0739 | 0.0927 | 0.0998 |
| Distress num | 10 | 11 | 23 | 38 | 54 | 49 | 45 | 50 | 53 |
| LASSET | 3.3500 | 2.8800 | 2.9600 | 3.3400 | 3.3500 | 3.2400 | 3.2800 | 3.3100 | 3.2000 |
| CAP_ADE | 0.0759 | 0.0759 | 0.0843 | 0.0862 | 0.0908 | 0.0965 | 0.1031 | 0.1129 | 0.1219 |
| LOAN_RATIO | 0.6531 | 0.6520 | 0.4896 | 0.4051 | 0.4177 | 0.4319 | 0.4570 | 0.4374 | 0.4603 |
| LOAN_COV | 10.390 | 9.2900 | 1.7400 | 5.9500 | 4.4800 | 2.6400 | 0.6950 | 2.9000 | 2.2500 |
| LIQ | 0.0866 | 0.0836 | 0.2260 | 0.2863 | 0.3027 | 0.3318 | 0.3252 | 0.3503 | 0.3585 |
| RES_CAP | 0.1613 | 0.1138 | 0.1925 | 0.2383 | 0.2838 | 0.2830 | 0.2598 | 0.2939 | 0.3212 |
| OPE_EFF | 0.0190 | 0.0229 | 0.0275 | 0.0272 | 0.0308 | 0.0260 | 0.0288 | 0.0304 | 0.0358 |
| INC_CAP | 0.0373 | 0.0284 | 0.0337 | 0.0338 | 0.0382 | 0.0338 | 0.0355 | 0.0353 | 0.0325 |
| LAB_COST | 0.2140 | 0.2752 | 0.2708 | 0.2452 | 0.2427 | 0.2424 | 0.2431 | 0.2407 | 0.2807 |
| MEM_GRO | 0.0159 | −0.0275 | −0.0092 | 0.0012 | 0.0134 | 0.0010 | −0.0072 | −0.0066 | −0.0224 |
| LOAN_GRO | 0.0379 | −0.0930 | −0.0143 | −0.0521 | −0.0332 | 0.0590 | 0.0285 | 0.0502 | −0.1007 |
| SHARE_GRO | 0.0346 | −0.0497 | 0.0078 | 0.0529 | 0.0197 | −0.0105 | 0.0085 | −0.0071 | −0.0185 |
| STO_RET | 0.0166 | 0.0134 | 0.0072 | 0.0086 | 0.0098 | 0.0059 | 0.0077 | 0.0068 | 0.0047 |
| ROA | 0.0045 | 0.0055 | 0.0055 | 0.0057 | 0.0035 | 0.0060 | 0.0062 | 0.0049 | −0.0033 |
| M1b | 0.1058 | −0.0102 | 0.1701 | 0.1183 | 0.1898 | 0.0710 | 0.0530 | 0.0644 | −0.0294 |
| GDP | −0.0249 | 0.0487 | 0.0267 | 0.0638 | 0.0324 | 0.0432 | 0.0541 | −0.0135 | −0.0147 |
| RATE | 0.0462 | 0.0409 | 0.0238 | 0.0147 | 0.0111 | 0.0122 | 0.0140 | 0.0153 | 0.0171 |
| RATE_SPR | 0.0299 | 0.0290 | 0.0315 | 0.0263 | 0.0255 | 0.0245 | 0.0209 | 0.0183 | −0.0176 |
| π_merton | 0.2015 | 0.1116 | 0.1566 | 0.1775 | 0.1287 | 0.1575 | 0.1603 | 0.1437 | 0.1332 |
| π_shumway | 0.2542 | 0.1892 | 0.2832 | 0.2778 | 0.2609 | 0.2591 | 0.2660 | 0.2636 | 0.2322 |
| σv_merton | 0.4142 | 0.2191 | 0.5236 | 0.5067 | 0.3432 | 0.4642 | 1.4769 | 0.3482 | 0.3420 |
| σv_shumway | 0.3551 | 0.2637 | 0.4107 | 0.3961 | 0.3486 | 0.3887 | 0.8067 | 0.3828 | 0.3633 |
| π_α_up | 0.4450 | 0.4136 | 0.5204 | 0.5066 | 0.4991 | 0.4835 | 0.5034 | 0.4808 | 0.4721 |
| π_α_down | 0.2302 | 0.1568 | 0.2441 | 0.2437 | 0.2221 | 0.2250 | 0.2309 | 0.2308 | 0.2001 |
| π_β_up | 0.6121 | 0.5518 | 0.6852 | 0.6703 | 0.6565 | 0.6384 | 0.6483 | 0.6575 | 0.6050 |
| π_β_down | 0.1525 | 0.0990 | 0.1599 | 0.1617 | 0.1402 | 0.1479 | 0.1594 | 0.1492 | 0.1321 |

Note: 1. This table points out the sample means of time-varying covariates from 2001 to 2009, including the data of all the credit unions that were under normal operation and those facing financial distress. 2. It takes the π_shumway according to the setting value of term structure and equity volatility and develops four simplified probabilities. Those simplified probabilities define π(term structure volatility, equity volatility), such as π_α_up (0.5,0.25), π_α_down (0.005,0.25), π_β_up (0.05,2.5), π_β_down (0.05,0.025).

The relationship between asset size and performance is widely documented in the theoretical and empirical banking literature. The factors make large financial institutions have lower costs and higher requirements, from economies of scale and loan grants to monitoring systems. Accordingly, the relationship between asset scale and performance means that small-scale credit unions own higher risks of confronting financial distress than larger-scale institutions. Table 6 shows a negative coefficient on LASSET in the hazard functions and Table 4 indicates a slow increase in the mean assets of credit unions throughout the sample period. Additionally, Table 5 shows that the financial distress credit unions are considerably smaller on average than those that do not have financial distress.

Age might represent some managerial implications that could impact the probability of financial distress, and we find specific concerns about the coefficient on LAGE. Table 3 shows there is little difference between the age, distribution by year of formation, of the sample as a whole and the credit unions that have financial distress. The financially distressed credit unions were mainly established between 1971 and 1980, but between 1991 and 2000, the number of credit unions founded had increased.

Table 5 points out the sample means of time-varying covariates from 2001 to 2009, including the data of all the credit unions that were under normal operation and those facing financial distress. We might expect either a positive or a negative relationship between CAP_ADE and the probability of financial distress. A positive relationship might be expected if a credit union holds excess capital because it has limited opportunities for growth. It means a highly capitalized credit union might not focus on its main business. Alternatively, it might be poorly capitalized, and seeking to improve its loan business proportion of assets. The summary statistics indicate that the average value of CAP_ADE is slightly higher for financial distress credit unions than for the normal sample.

Loans are typically less liquid and riskier than other assets; a credit union with a low loans-to-assets ratio, a low overdue reserves-to-nonperforming loans ratio, or a high loan growth rate might be at greater risk of financial distress. So, we would expect a negative coefficient on LOAN_RATIO and LOAN_COV; a positive coefficient on LOAN_GRO in the hazard function. Alternatively, credit unions with relatively small loan portfolios or low overdue reserves for non-performing loans might reduce interest income or increase loss reserves. The higher loan growth rate might cause a higher delinquency rate. The summary statistics indicate that the average values of LOAN_RATIO and LOAN_COV are generally lower for financial distress credit unions than for the normal sample, but the average values of LOAN_GRO are exactly the opposite.

A highly liquid credit union might be at lower risk than an illiquid one because high liquidity means a high proportion of current assets, or because it could settle the financial distress situation itself. So, we expect a negative coefficient on LIQ. According to the summary statistics, the average LIQ of financial distress credit unions is lower than the average for the normal sample.

A high RES_CAP represents the degree of risk tolerance when handling loan grants, and the results show a significant positive relationship. In the circumstance that a loan business continues to shrink, credit unions with a low reserve capacity for loan ratios elevate the ratio, and credit unions depending on a loan interest margin face a rapid drop in profitability. Although loan default risk tolerance has increased, this has increased the chances of encountering financial distress.

For management, a measurement of cost efficiency in an organization depends on how much of the total assets are taken up by operation fees. Credit Unions with little efficiency usually tend to face financial distress, and the operating expense ratio (OPE_EFF) has a close relationship to hazard functions, which indicates that the higher the operating expense ratio, the higher the risk of financial distress. The ratio of personnel expense also has a highly significant positive relationship to hazard functions, which means that the higher the labor cost ratio (LAB_COST), the higher the risk of financial distress. Even though the ratio of income capacity (INC_CAP) presents an inverse relationship to hazard

functions, indicating a higher risk with a lower ratio, this ratio is significant only to credit unions with a single common bond in Table 6.

**Table 6.** Hazard function estimation results.

| Equation | I | II | III | IV | V | VI | VII | VIII |
|---|---|---|---|---|---|---|---|---|
| Sample | 01~03 | 04~06 | 07~09 | All | EAST | WEST | Single | MULT |
| $\pi$_merton | 0.043 | −8.522 | 0.712 | −1.424 | −5.408 | −2.146 ** | −5.527 | −0.274 |
|  | (0.07) | (−0.90) | (0.15) | (−1.48) | (−0.90) | (−2.18) | (−1.50) | (−0.26) |
| $\pi$_shumway | −0.411 | 5.286 | 28.338 * | 5.276 | 16.165 | 1.154 | −1.703 | 5.830 |
|  | (-0.12) | (0.49) | (1.87) | (1.19) | (1.14) | (0.25) | (-0.18) | (1.14) |
| $\sigma$v_merton |  |  |  | −0.449 | 4.297 | −0.575 ** | −0.569 | 0.080 |
|  |  |  |  | (−1.54) | (1.24) | (−1.98) | (−0.55) | (0.07) |
| $\sigma$v_shumway | −0.459 | −0.397 | 1.922 | 1.785 ** | −18.14 ** | 2.264 *** | 1.352 | 0.209 |
|  | (−0.54) | (−0.25) | (1.87) | (2.21) | (−2.71) | (2.79) | (0.66) | (0.53) |
| $\pi$_$\alpha$_up | −0.796 | −3.812 | −3.339 | −3.454 ** | −15.76 *** | −3.361 ** | −7.712 | −4.729 *** |
|  | (−1.06) | (−0.64) | (−0.50) | (−2.45) | (−3.23) | (−2.26) | (−1.59) | (−3.16) |
| $\pi$_$\alpha$_down | −1.334 | 0.593 | −13.201 | −3.191 | −30.98 * | 0.892 | 11.072 | −3.835 |
|  | (−0.39) | (0.05) | (−0.79) | (−0.74) | (−1.71) | (0.20) | (1.14) | (−0.77) |
| $\pi$_$\beta$_up | 0.778 | 1.049 | −1.756 | 0.737 | 7.695 * | 1.863 * | 1.060 | 1.375 |
|  | (0.91) | (0.22) | (0.56) | (0.71) | (1.92) | (1.69) | (0.35) | (1.21) |
| $\pi$_$\beta$_down | 7.222 *** | 1.873 | −17.406 | 3.214 * | 49.11 *** | 2.145 | 1.307 | 2.694 |
|  | (3.52) | (0.76) | (−0.75) | (1.70) | (2.95) | (1.14) | (0.19) | (1.26) |
| LAGE | −0.51 | 1.426 | 5.078 *** | 0.870 ** | 0.954 | 1.269 ** | 0.189 | 1.502 *** |
|  | (−1.20) | (0.92) | (3.06) | (2.35) | (1.33) | (2.54) | (0.29) | (3.25) |
| LASSET | 0.132 | −1.646 | −14.078 *** | −1.514 *** | −3.172 *** | −1.287 *** | −0.413 | −2.892 *** |
|  | (1.31) | (−1.35) | (−6.46) | (−6.52) | (−3.55) | (−5.31) | (−1.14) | (−8.84) |
| CAP_ADE | 0.506 | −5.078 | 5.098 | −7.97 ** | −14.204 | −5.956* | −0.572 | −15.036 *** |
|  | (0.19) | (−0.59) | (0.39) | (−2.46) | (−1.62) | (−1.68) | (−0.07) | (−4.06) |
| LOAN_RATIO | 0.406 | −1.975 | −6.371 ** | −0.542 | −2.590 | −0.483 | −0.633 | −0.250 |
|  | (1.04) | (−1.07) | (−2.36) | (−0.84) | (−1.13) | (−0.70) | (−0.415) | (−0.34) |
| LOAN_COV | −0.001 | −0.0045 | 0.0026 | −0.0002 | 0.0015 | −0.0004 | −0.0026 | −0.0007 |
|  | (−0.78) | (−1.14) | (0.32) | (−0.13) | (0.22) | (−0.19) | (−0.79) | (−0.24) |
| LIQ | −0.617 ** | −1.270 | −2.275 | −1.393 *** | −2.163 | −1.226 ** | −1.111 | −1.299 ** |
|  | (−2.35) | (−1.06) | (−1.16) | (−2.74) | (−1.29) | (−2.29) | (−0.97) | (−2.28) |
| RES_CAP | −0.115 | 1.792 | −2.675 | 1.990 *** | 0.707 | 1.985 *** | 2.907 * | 1.496 ** |
|  | (−0.18) | (1.59) | (−1.29) | (3.55) | (0.19) | (3.50) | (1.71) | (2.48) |
| OPE_EFF | −1.175 | −1.520 | −14.327 | 8.718 ** | 0.676 | 8.072 * | 50.877 *** | 7.567 ** |
|  | (−0.83) | (−0.11) | (−1.12) | (2.53) | (0.12) | (1.72) | (3.15) | (2.12) |
| INC_CAP | 0.322 | −8.869 | 3.470 | −2.239 | 3.170 | −2.210 | −48.004 *** | −0.290 |
|  | (0.47) | (−1.02) | (0.30) | (−1.14) | (0.32) | (−1.05) | (−3.62) | (−0.15) |
| LAB_COST | 1.180 *** | 3.329 *** | 13.182 *** | 7.312 *** | 14.235 *** | 5.873 *** | 2.248 | 7.892 *** |
|  | (2.76) | (3.02) | (5.33) | (11.53) | (6.29) | (8.77) | (1.63) | (10.88) |
| MEM_GRO | 0.085 | 1.497 | −3.324 | −0.523 | −0.588 | −0.514 | −2.2696 * | 0.114 |
|  | (0.34) | (1.26) | (−1.31) | (−0.87) | (−0.39) | (−0.78) | (−1.89) | (0.16) |
| LOAN_GRO | 0.029 | 0.713 * | −0.087 | 0.216 | 0.544 | 0.249 * | 0.159 | 0.072 |
|  | (0.18) | (1.79) | (−0.16) | (1.51) | (0.76) | (1.70) | (0.88) | (0.30) |
| SHARE_GRO | −0.614 ** | 0.370 | 7.236 *** | 1.112 ** | 1.094 | 1.310 ** | 1.496 * | 1.075 |
|  | (−2.00) | (0.41) | (3.18) | (2.10) | (0.74) | (2.26) | (1.74) | (1.53) |
| ROA | 0.420 | −21.892 *** | −32.769 *** | −9.802 *** | −6.692 | −8.193 *** | −2.245 | −14.012 *** |
|  | (0.49) | (−4.25) | (−3.10) | (−4.97) | (−1.34) | (−3.70) | (−0.50) | (−6.22) |
| M1b | 0.906 ** | −1.741 | 3.489 | 0.313 | −1.756 | 1.177 | −4.073 | 1.272 |
|  | (2.51) | (1.25) | (1.55) | (0.30) | (-0.68) | (1.01) | (−1.53) | (1.11) |
| GDP | 2.365 ** | −12.656 * | −2.316 | −1.599 | −6.502 | 0.062 | −9.384 ** | −0.099 |
|  | (2.54) | (−1.77) | (−0.74) | (−0.85) | (−1.42) | (0.03) | (−2.03) | (−0.05) |
| RATE |  |  |  | 5.464 | −9.992 | 12.754 | −10.608 | 5.571 |
|  |  |  |  | (0.62) | (−0.46) | (1.28) | (−0.47) | (0.57) |
| RATE_SPR |  |  |  | −68.56 *** | −34.799 | −71.190 *** | 56.11 | −104.52 *** |
|  |  |  |  | (−3.04) | (−0.62) | (−2.90) | (1.03) | (−4.24) |
| Observations | 922 | 929 | 941 | 2792 | 520 | 2272 | 457 | 2335 |
| Credit unions | 308 | 311 | 314 | 314 | 58 | 256 | 53 | 261 |
| distress | 44 | 141 | 148 | 333 | 61 | 272 | 57 | 276 |
| Adjusted R-squared | 0.2411 | 0.4874 | 0.5145 | 0.3688 | 0.4177 | 0.3683 | 0.425 | 0.3756 |

Note: 1. This table includes the default estimation of each model and its adjusted R-squared. This research adopted pooled panel data to estimate the fixed sample effects, and the Hausman test was also used to examine the random and fixed effects of the models. Since the results were all significant, this study chose the fixed effects method to conduct the estimation. 2. This table reports the hazard function estimation results. Equations I–III are based on credit union data collected from 2001 to 2003, 2004 to 2006, and 2007 to 2009. In Equation III, hazard function estimates indicate that the financial tsunami happened between 2007 and 2009. In Equation IV, hazard function estimates are provided for every situation of all the covariates. Equations V and VI repeat the estimation in Equation IV, using only the data for eastern and western Taiwan credit unions, respectively. Finally, Equations VII and VIII repeat the estimation in Equation IV, using only the data for single and multiple common bond credit unions, respectively. 3. A positive coefficient on a particular variable implies that the hazard rate is increasing in that variable, or that the expected time to default is decreasing in that variable. Standard errors are in parentheses (*** Estimated coefficient significantly different from zero, two-tail test, 1% significance level. ** As above, 5% significance level. * As above, 10% significance level).

In terms of the factors, including the number of members and money paid for shares, the increase in the number of members is significant only to credit unions with a single common bond. The growth of the money paid for shares shows an insignificant positive relationship to credit unions with a single common bond and in western areas, which shows that a higher growth rate is accompanied by increased pressure for capital utilization; if capital has not been properly utilized, the chances of having financial distress will increase. It seems that credit unions with poor profitability are more possibly to have financial distress than those with high profitability; so, we expect a negative coefficient on ROA in the hazard function. It shows that the average ROA of financial distress credit unions is always lower than the average ROA of the normal sample.

For overall exterior environmental factors, only the spreads of deposits and loans in the banking business show a significant inverse relationship to hazard functions: the smaller the spreads, the higher the risk of financial distress. With such convenient lending and financing channels, lenders can choose banks instead of credit unions, and this situation will make the loan business in credit unions shrink and its overall earnings drop. The other three overall variables have no significant relationship to hazard functions. Table 5 shows that the financial tsunami struck in 2008 and 2009. During this period, the annual growth of the average interest rate of deposits (RATE) reached a historical low, as well as the spreads of deposits and loans (RATE_SPR), GDP, and M1b. In addition, the rate of financial distress increased.

Finally, only the individuals who have the same common bond (membership) of the credit union can use the credit union's services. In the hazard functions, the dummy variable MULT distinguishes between single and multiple common bond credit unions, and the dummy variable EAST does the same for eastern and western credit unions of Taiwan. The summary statistics suggest that a relatively high proportion of financial distress credit unions are single common bond, but the proportions of financial distress that are eastern and western credit unions are similar to those for the sample as a whole.

## 5. Empirical Evidence and Results

### 5.1. Hazard Function Estimation Results

The empirical results are shown in Tables 6 and 7, including the default estimation of each model and its adjusted R-squared. This research adopted pooled panel data to estimate the fixed sample effects, and the Hausman test was also used to examine the random and fixed effects of the models. Since the results were all significant, this study chose the fixed effects method to conduct the estimation.

Table 6 reports the hazard function estimation results. Equations I–III are based on credit union data collected from 2001 to 2003, 2004 to 2006, and 2007 to 2009. In Equation III, hazard function estimates indicate that the financial tsunami happened between 2007 and 2009. The seven variables showing its significance indicate that the crisis was much greater in this period than in the 2001 to 2006 time period. Equation III possesses better explanation ability than Equations I and II, and, interestingly, most EDF values in Equations I–III are insignificant.

In Equation IV, hazard function estimates are provided for every situation of all the covariates. Equations V and VI repeat the estimation in Equation IV, using only the data for eastern and western Taiwan credit unions, respectively. Finally, Equations VII and VIII repeat the estimation in Equation IV, using only the data for single and multiple common bond credit unions, respectively. In Equation IV, the coefficient of $\pi\_\beta\_down$ is statistically significant, opposite to $\pi\_shumway$ and $\pi\_merton$. The results also show that ROA is statistically significant, but the coefficient of $\pi\_merton$ is insignificant, even though it adopts the same financial data. The insignificance of $\pi\_merton$ indicates that it is unnecessary to determine the functional form of sophisticated equations. In effect, there is no need to calculate the asset value and volatility of credit unions in the Merton DD model.

**Table 7.** Models' estimation results and forecasts.

| Models | I | II | III | IV | V | VI | VII | VIII |
|---|---|---|---|---|---|---|---|---|
| Methods | Merton | Shumway | Merton & Shumway | Hazard & Merton | Hazard | Hazard & Shumway | Hazard & EDF | Logit |
| $\pi$_merton | 1.264 *** | | −4.951 *** | 1.104 ** | | | −1.685 ** | 0.500 |
| | (4.18) | | (−5.8) | (1.98) | | | (−2.09) | (0.37) |
| $\pi$_shumway | | 2.563 ** | 4.461 *** | | | 1.135 | 1.776 | 3.343 |
| | | (2.13) | (3.59) | | | (1.00) | (1.52) | (1.26) |
| $\pi$_$\beta$_down | | 6.086 *** | 9.157 *** | | | 2.92 * | 4.053 ** | −3.536 |
| | | (3.47) | (5.03) | | | (1.79) | (2.36) | (−0.90) |
| z-score | | | −0.005 | | | −0.007 * | −0.008 * | −0.036 * |
| | | | (−1.14) | | | (−1.91) | (−1.95) | (−1.70) |
| LAGE | | | | 0.714 ** | 0.716 ** | 0.645 * | 0.602 * | 0.012 |
| | | | | (2.10) | (2.11) | (1.90) | (1.77) | (0.03) |
| LASSET | | | | −1.497 *** | −1.502 *** | −1.471 *** | −1.459 *** | −0.312 ** |
| | | | | (−6.54) | (−6.55) | (−6.44) | (−6.39) | (−2.03) |
| CAP_ADE | | | | −5.791 ** | −6.197 ** | −5.108 * | −4.958 * | 0.769 |
| | | | | (−2.04) | (−2.19) | (−1.77) | (−1.72) | (0.15) |
| LIQ | | | | −1.039 *** | −1.030 *** | −1.029 *** | −1.047 *** | −2.696 *** |
| | | | | (−2.71) | (−2.60) | (−2.67) | (−2.72) | (−3.07) |
| RES_CAP | | | | 2.336 *** | 2.337 *** | 2.227 *** | 2.180 *** | 1.655 |
| | | | | (4.59) | (4.59) | (4.39) | (4.29) | (1.36) |
| OPE_EFF | | | | 6.866 ** | 7.143 ** | 6.404 ** | 6.506 ** | −1.341 |
| | | | | (2.31) | (2.41) | (2.17) | (2.21) | (−0.09) |
| LAB_COST | | | | 7.658 *** | 7.673 *** | 7.414 *** | 7.317 *** | −1.893 |
| | | | | (12.46) | (12.48) | (12.06) | (11.88) | (−1.18) |
| SHARE_GRO | | | | 1.065 ** | 1.086 ** | 0.966 ** | 0.946 ** | −1.047 |
| | | | | (2.29) | (2.33) | (2.08) | (2.04) | (−0.53) |
| ROA | | | | −10.888 *** | −10.925 *** | −10.167 *** | −9.820 *** | −686.3 *** |
| | | | | (−6.02) | (−6.04) | (−5.61) | (−5.40) | (−13.94) |
| RATE_SPR | | | | −71.25 *** | −73.39 *** | −66.97 *** | −66.60 *** | 149.19 *** |
| | | | | (−5.92) | (−6.12) | (−5.51) | (−5.48) | (5.02) |
| Observations | 2793 | 2793 | 2973 | 2793 | 2793 | 2793 | 2793 | 2793 |
| Hausman test | 1.82(0.17) | 53.08(0.00) | 67.00(0.00) | 69.35(0.00) | 70.55(0.00) | 81.31(0.00) | 82.85(0.00) | |
| Adjusted R-squared | 0.0058 | 0.2404 | 0.2502 | 0.3613 | 0.3605 | 0.3664 | 0.3672 | |
| F-statistic | 17.53 | 3.80 | 3.93 | 5.87 | 5.87 | 5.95 | 5.95 | |
| RMSE | 2.7551 | 2.5649 | 2.5475 | 2.2666 | 2.2689 | 2.2644 | 2.2622 | - |

Note: 1. This table shows the estimation and forecast results of empirical models, including the coefficient estimation on each variable, and the Adjusted R-squared, F test, and Root Mean Squared Error (RMSE) of models I to VIII. The 2008 to 2009 time period was chosen for the out-of-sample forecasts. The RMSE values were obtained to compare the actual and expected value of each model and further assess its predictive capability. 2. Models I and II are the estimations of single variable hazard models, including the explanation for the time-varying $\pi$_merton and $\pi$_shumway estimation. In Model III, hazard models simultaneously estimate $\pi$_merton, $\pi$_shumway, z-score, and the simplified probability estimation values. When coordinating out-of-sample forecast models with financial and organizational variables, the research shows $\pi$_merton and $\pi$_shumway as model IV and model VI. In model VII, only $\pi$_$\beta$_down shows significance, and $\pi$_shumway is also insignificant. Model VIII is a logit model that considers all the variables, and it is apparent that there are more significant variables in the hazard model than in the logit model. 3. A positive coefficient on a particular variable implies that the hazard rate is increasing in that variable, or that the expected time to default is decreasing in that variable. Standard errors are in parentheses (*** Estimated coefficient significantly different from zero, two-tail test, 1% significance level. ** As above, 5% significance level). * As above, 10% significance level. 4. RMSE(Root Mean Squared Error): The method of calculating forecast accuracy. The time period chosen to conduct out-of-sample forecasts is between 2008 and 2009, comparing the actual and expected value of each model and further assessing their predictive capability.

The coefficients of $\pi$_merton, $\pi$_shumway, $\pi$_$\alpha$_down, and $\pi$_$\beta$_up in Equations I through VIII are mostly insignificant, while the coefficient on $\pi$_$\beta$_down is positive and significant in Equations I, IV, and V. The coefficient on $\pi$_$\alpha$_up is negative and significance in Equations IV, V, VI, and VIII. It indicates that the higher is the credit union's default, the more likely is the credit union to have financial distress. In addition, the results show that the proper probability setting for credit unions is different from the assumption of $\pi$_shumway, which still has space to revise down the coefficient of equity volatility. This finding indicates that the purpose of the Merton DD model is to obtain a functional form instead of a basic solution form.

The anticipated inverse relationship between asset size and the hazard of financial distress is evident in most of the hazard function estimations reported in Table 6. The coefficients on LASSET are negative and strongly significant coefficients in Equations III, IV, V, VI, and VIII; therefore, a subdivision of the sample by period, by located region, or by common bond does not affect this strong underlying relationship between size and the hazard of financial distress. The significant coefficient on LAGE in Equations III, IV, VI, and VIII suggests that older credit unions are at greater risk of financial distress. This pattern is repeated in most of the other estimations, although not all of the coefficients are significant.

As for asset quality, the coefficients on CAP_ADE generally lend support to the explanations for a negative relationship in Equations IV, VI, and VIII between CAP_ADE and the hazard of financial distress, which indicates that the lower the capital adequacy ratio, the higher the risk. It is similar to Pille and Paradi's (2002) conclusion that a simple equity/asset ratio is a good predictor of failure. The significant coefficients are the coefficient of the western area and multiple common bond credit unions. Surprisingly, the coefficient on LOAN_RATIO is negative and significant only in Equation III and the coefficients of LOAN_COV are insignificant; however, Tables 4 and 5 show that financially distressed credit unions tend to have a lower LOAN_RATIO and LOAN_COV ratio.

For financial risk, the liquidity ratio LIQ is not shown to be significant for credit unions with a single common bond, credit unions in eastern areas, or the 2004 to 2009 period. While credit unions have a lower liquidity ratio, they might have financial distress. The coefficients of RES_CAP show positive and significant relationships in Equations IV VI, VII, and VIII. It is paradoxical that credit unions with a higher reserve to total loan ratio might have financial distress. The possible reason is loan ratio decreased gradually in the last decade.

For management efficiency, the coefficients of OPE_EFF show positive and significant relationships in Equations IV, VI, VII, and VIII, which means that credit unions with higher operating expenses to total assets ratio might have financial distress. The coefficients of LAB_COST all have a highly significant positive relationship except Equations VII. The higher the salaries to total income ratio, the more likely that financial distress might happen. The coefficient of INC_CAP shows a significant inverse relationship only in credit unions with a single common bond.

For business growth, the coefficients of MEM_GRO and LOAN_GRO present insignificant relationships in most equations; however, the coefficients of SHARE_GRO show positive and significant relationships in Equations I, III, IV, VI, and VII that mean credit unions with the higher share growth rate might have financial distress. The anticipated inverse relationship between profitability and the hazard of financial distress is evident throughout Table 6. The coefficients on ROA are negative and significant in all estimations except Equations I, V, and VII.

At last, the coefficients of M1b, GDP, and RATE have no significant relationship in most Equations; however, the coefficients of RATE_SPR have a significant inverse relationship in Equations except for the eastern areas and credit unions with a single common bond, which means credit unions with the lower spreads from deposit and loan of local banks might have financial distress.

### 5.2. Out-of-Sample Forecasts

Table 7 shows the estimation and forecast results of empirical models, including the coefficient estimation on each variable, and the Adjusted R-squared, F test, and Root Mean Squared Error (RMSE) of models I to VIII. The 2008 to 2009 time period was chosen for the out-of-sample forecasts. The RMSE values were obtained to compare the actual and expected value of each model and further assess its predictive capability.

Models I and II are the estimations of single variable hazard models, including the explanation for the time-varying $\pi$_merton and $\pi$_shumway estimation. The models show that $\pi$_merton and $\pi$_shumway are variables with statistical significance, especially the simplified probability values, $\pi$_$\beta$_down, in Equation II, which is based on the setting

of π_shumway that is modified by term structure volatility and equity volatility in time periods. The coefficient and standard deviation of π_β_down (0.05, 0.025) are especially closer to the estimation of π_shumway, which means that the equity volatility of credit unions is lower than the stock volatility of general companies; the result also qualifies the cooperative characteristics of credit unions. In Equation III, hazard models simultaneously estimate π_merton, π_shumway, z-score, and the simplified probability estimation values. Although π_merton is significant, it does not match the assumption that a high π_merton has a high possibility of financial distress. This means the setting of the expected default frequency of credit unions could adjust the equity volatility coefficient of π_shumway and reducing the value of π_β_down could make Equations II and III significant. Then, a positive relationship with the highest intensity would be presented, which matches the assumption that high default probability also has high risk. Hence, it is concluded that π_merton is not appropriately used to predict the default probability of Taiwan credit unions.

When coordinating out-of-sample forecast models with financial and organizational variables, the research shows π_merton and π_shumway as model IV and model VI. The π_shumway in model VI is statistically insignificant, and the π_β_down in model VI is statistically significant. In model VII, only π_β_down shows significance, and π_shumway is also insignificant. In addition, the predictive capability of model VII is more significant than models I, II, and III, which are the models that simply take account of the default probability of market equity volatility. The result matches the research performed by Vassalou and Xing (2004) and Bharath and Shumway (2008). The π_merton offers more than the predictive capability of market equity; it obviously also provides an instrument to measure probability and to construct a linear combination to calculate expected default frequency; however, π_β_down is more statistically significant than π_merton and π_shumway on hazard models and out-of-sample forecasts.

Model VIII is a logit model that considers all the variables, and it is apparent that there are more significant variables in the hazard model than in the logit model; this result also matches the conclusion proposed by previous research.

We find the adjusted R-squared of model V (Hazard) is 36.05% in Table 7. Then we add EDFs in model VII (Hazard, EDF and π_β_down); the adjusted R-squared increases from 36.05% to 36.72%. The F-statistic of model V is 5.87, which is up to 5.95 in model VII. The statistic F-statistic is the hypothesis used to test the significance of the model; that is, the larger the F-statistic value, the more inclined to reject the null hypothesis that the model is not significant. The higher the explanatory power of the model explanatory variables, the smaller the unexplained variation. So, we agree that the EDF of Merton and Shumway can be regarded rather as a target variable than an explanatory variable when developing failure prediction models; however, we want to expand and explain the variables of the equation by Crapp and Stevenson (1987) and Goddard et al. (2009).

The time period chosen to conduct out-of-sample forecasts is between 2008 and 2009, comparing the actual and expected value of each model and further assessing their predictive capability. Evaluate the predictive ability of the model by comparing out-of-sample actual and predicted values. We find the RMSE of model V (Hazard) is 2.2689 in Table 7. The value is higher than the RMSE (2.2666) of model IV or the RMSE (2.2644) of model VI. Then, the RMSE is 2.2622 in model VII (Hazard, EDF, and π_β_down); therefore, the research shows that model VII has better results than the other models regardless of the estimated number of significant variables, the explanation of models, and the out-of-sample forecast.

Finally, we find the robust tests that the coefficients on LASSET are negative and strongly significant coefficients in models IV, V, VI, VII, and VIII. The relationship between asset scale and performance means that small-scale credit unions own higher risks of confronting financial distress than larger-scale institutions. The significant coefficient on LAGE in models IV, V, VI, and VII suggests that older credit unions are at greater risk of financial distress. The coefficients on CAP_ADE generally lend support to the explanations

for a negative relationship in models IV, V, VI, and VII between CAP_ADE and the hazard of financial distress. Credit unions with low capitalization are at greater risk of financial distress. This could be that poorly capitalized credit unions have been inefficiently managed and expanded assets rapidly. The liquidity ratio LIQ is shown to be significant to credit unions in all models. Lowly liquid credit unions might have financial distress, perhaps because of the approachability of their assets in non-liquid form, or their risky and fixed assets. The coefficients of RES_CAP, OPE_EFF, and LAB_COST all show positive and significant relationships in all models that mean credit unions with the higher ratio might have financial distress. Operating expenses to total assets rate, salaries to total income rate mean the operating burden of credit unions. Then the coefficients of SHARE_GRO also show positive and significant relationships in models IV, V, VI, and VII. Credit unions that have a high share growth rate might be more likely to have financial distress without fund allocation performance. The coefficients of ROA are negative and significant in all models. At last, the coefficients of RATE_SPR are negative and significant in models IV, V, VI, and VII. The spreads from deposits and loans of local banks decreased and the rate of credit unions that have financial distress increased. The spreads from deposits and loans of local banks represent the degree of business competitiveness of credit unions. The smaller the spreads, the smaller the competitiveness of credit unions which are not conducive to promoting business.

## 6. Conclusions

In recent years, the Taiwan credit union movement has experienced a wave of financial distress; however, the increase in financial distress activity has remained largely unknown in the academic literature. In this study, we seek to fill this gap by examining the determinants of financial distress activity for the Taiwan credit union movement. Most previous studies find the factors leading to financial distress. An important contribution of this paper is the incorporation of Merton DD variables into a model of the determinants of the probability of financial distress for credit unions.

We examined the accuracy and contribution of the hazard model compared to other models. After conducting an empirical test of hazard models, our research discovers that $\pi$_merton does not possess significant predictive capability on default probability. Considering other default predictive variables will improve the $\pi$_merton; it is proposed that $\pi$_shumway, constructed in z-score form, can improve predictive capability and that the predictive capability can be improved even further if the strength of equity volatility can be adjusted to a lower status. For the predictive capability of out-of-samples, the performances of a uni-predictive index, such as $\pi$_merton and $\pi$_shumway, are poorer than the hazard models, which add $\pi$_$\beta$_down.

The study finds that there is a strong inverse relationship between profitability and the hazard of financial distress. The relationship between asset scale and performance means that small-scale credit unions own higher risks of confronting financial distress than larger-scale institutions. Lowly liquid credit unions might have financial distress, perhaps because of the approachability of their assets in non-liquid form, or their risky and fixed assets. Credit unions with low capitalization are at greater risk of financial distress. This could be that poorly capitalized credit unions have been inefficiently managed and expanded their assets too rapidly. In addition, the spreads from deposits and loans of local banks decreased and the rate of credit unions that have financial distress increased.

In managerial efficiency, we have found evidence of a strong positive relationship between operating expenses to total assets rate and the hazard of financial distress. In addition, we also found that the salaries to total income rate has the same relationship with financial distress. Credit unions that have a high share-growth rate might be more likely to have financial distress without fund allocation performance. Older credit unions are at greater risk of financial distress. The ability to more accurately predict the probability of financial distress and to determine the financial health of credit unions can help government regulators and policy makers in allocating resources and predicting crises.

Another area for future research is to focus on the differences in financial distress prediction between profit-making companies and cooperative organizations. The limitation of the research is that credit unions do not collect capital by issuing shares to the public; the capital is only the combined deposits of internal members. When defining the volatility of shareholder equity, one cannot collect the data according to the estimated condition of the original Merton model. Calculating the volatility requires replacing the stock price with net income per month divided by deposited shares.

Normally, profit-making companies have ratings from credit rating agencies; however, cooperative financial institutions do not have credit ratings from relative professional agencies and their financial operations normally remain private. If financial distress occurs in a cooperative financial institution, it is often irreparable. Verifications of the models mentioned above provide cooperative financial institutions with a common operating model to predict financial distress. Thus, different models with different characteristics are improved by the research data. Some models have a broader scope of application with smaller hypothesis restrictions. The Merton DD model concerns the real-time market data; the logit model has practicability and convenience; the hazard model considers changes according to time and predicts the time point of default. For profit-making companies and cooperative organizations, different organizational characteristics will suit different models. A financial distress prediction model that can prevent credit unions from facing financial distress will combine the hazard model with a corrected Merton DD. This combined model can reflect market volatility information in real-time and predict the time point when distress occurs.

The findings of Boubaker et al. (2020) suggest that the adoption of corporate social responsibility (CSR) practices of US-listed firms comes with less distress and default risks, likely leading to a more attractive corporate environment, better financial stability, and more crisis-resilient economies. Lizares and Bautista (2021) empirically analyze the usefulness of combining accounting and auditing data in order to predict corporate financial distress. Specifically, findings indicate that the number of disclosures included in the report, as well as disclosures related to going concern, firms' assets, and firms' recognition of revenues and expenses, contribute the most to the prediction. In the future, our research may examine how the CSR level affects financial distress risk in credit unions. Essentially, credit unions are not-for-profit organizations with a high degree of CSR. For managers of credit unions, the importance of the audit report disclosures is for anticipating a financial distress situation. For regulators and auditors of credit unions, the regulation is changing in order to increase auditor's transparency through a more informative audit report.

**Author Contributions:** Conceptualization, C.-M.K.; Data curation, C.-M.K.; Formal analysis, C.-M.K.; Funding acquisition, L.L.; Methodology, C.-M.K.; Project administration, C.-M.K.; Resources, M.-C.W.; Software, L.L.; Supervision, M.-C.W.; Validation, M.-C.W.; Visualization, L.L.; Writing—original draft, C.-M.K.; Writing—review and editing, L.L. All authors have read and agreed to the published version of the manuscript.

**Funding:** This research received no external funding.

**Data Availability Statement:** The credit union balance sheet and income statement unique internal data are compiled from the annual reports published by the League of Credit Unions in Taiwan (CULROC).

**Acknowledgments:** The authors thank the editor and three anonymous referees for their helpful comments and suggestions.

**Conflicts of Interest:** The authors declare no conflict of interest.

## Appendix A

*The Merton DD Model*

Merton uses the option pricing theory presented by Black and Scholes (1973) and applies the theory to the measurement of credit risks. He considers the enterprise leverage

as equivalent to the shareholders' long call to the creditor. The call index of underlying assets is the company's asset value, and the exercise price is the liability. When liability is due, if the market value of the corporation's assets is lower than the liability value, shareholders will choose default, and the probability that the asset will not pay off the liability is called the probability of default. The Merton model uses the given equity market value and its variations to estimate corporation asset value and volatility, and then evaluates a company's probability of default and distance-to-default.

The distance-to-default calculates the closeness between a company's assets and the default point, which refers to a company's asset value benchmark at the time of default. If the company's asset value is lower than its liabilities when liabilities are due, the company is in default; however, according to KMV, the company still has the option to refinance, so the default will not happen immediately at the time when the asset value is lower than the book value of the liability. According to KMV's observation, the real default point situates between total liability and current liability. In other words, the company will default when its asset value is lower than its default point. To measure the risk of default, KMV combines three determining factors (asset value, asset risk, and leverage), which affect the probability of default, into one measurement variable: Distance to default (DD).

The DD indicates the distance between a company's asset value and default point and uses asset volatility to measure and standardize the distance; in other words, the standard deviation number between a company's asset value and default point, and the standardized distance-to-default helps a company to compare itself with others. A larger standardized number represents an asset value further away from the default point and an expected default frequency (EDF) that is smaller.

The Merton DD model uses the classical bond pricing model created by Merton (1974) to estimate the market value of liability. The Merton model has two important assumptions: The first is that the total value of a company must comply with the geometric Brownian motion, which is calculated as:

$$dV = \mu V dt + \sigma_v V dW \tag{A1}$$

where $V$ is the company's total value; $\mu$ is the expected continuously compounded return on $V$; $\sigma_v$ is the volatility of the company's asset value; $dW$ complies with the standard Wiener process.

The second assumption of the Merton model is that a company issues a discount bond, which is due on time T. Under those two assumptions, shareholder equity is a call option on the underlying value of a company with a strike price equal to the face value of the company's liability with due date T. Thus, the formula for shareholder equity and company value is called the Black–Scholes–Merton Formula. In the put-call parity, the value of company liability is equivalent to the value of a risk-free discount bond minus the value of a put option written on company assets, and a strike price equal to the face value of liability with due date T.

The company shareholder equity value in the Merton model complies with the formula below:

$$E = VN(d_1) - e^{-rT}FN(d_2) \tag{A2}$$

$E$ is the market value of the company's shareholder equity; $F$ is the face value of company liability; $r$ is the synchronized risk-free rate; $N(.)$ is the cumulative standard normal distribution function; $d_1$ is the formula below; $d_2$ is just $d_1 - \sigma_v\sqrt{T}$

$$d_1 = \frac{ln(V/F) + (r + 0.5\sigma_v^2)T}{\sigma_v\sqrt{T}} \tag{A3}$$

The Merton DD model makes use of two important formulas: The first is the Black–Scholes–Merton formula (A2), indicating the value of a company's shareholder equity as a function of company value. The second formula relates to the volatility of company value as

compared to the volatility of shareholder equity. Under Merton's assumption, shareholder equity is a function of company value and time; thus, the assumption positively complies with Ito's lemma:

$$\sigma_E = \left(\frac{V}{E}\right)\frac{\partial E}{\partial V}\sigma_v \tag{A4}$$

In the Black–Scholes–Merton model, it is certified that $\frac{\partial E}{\partial V} = N(d_l)$ nd the model complies with the assumption of the Merton model, in which the volatilities of company and shareholder equity are indicated as:

$$\sigma_E = \left(\frac{V}{E}\right)N(d_l)\sigma_v \tag{A5}$$

and $d_l$ is defined in formula (A3).

The Merton DD model basically uses two linear Equations (A2) and (A5) and transfers the value and volatility of a company's shareholder equity into the implied probability of default. For most applications, the Black–Scholes–Merton model describes functions of strike price, due date, underlying asset value, and risk-free rate, all of which are unable to observe value by option; also, one volatility variable still needs evaluation. In the Merton model, the value of the option can be observed from the value of company equity, but the underlying value of company assets is unobservable; therefore, the fact that V infers E is observed from market data (the number of the company's outstanding shares multiplied by current stock price). Similarly, the equity of volatility needs to be estimated in the Merton DD model, and the volatility of the underlying asset price must be calculated.

The first step in implementing the Merton DD model is to estimate $\sigma_E$ from the data of historical stock returns or optional implied volatility. The second step is to choose the prediction period and measure the face value of company liability. For example, using historical stock returns data to estimate the equity volatility, assume the prediction period is one year and make the book value of total company liability equal to the face value of company liability. The third step is to collect risk-free rate data and the market price of company equity. Except for V and $\sigma_v$, the values of various variables for formulas (A2) and (A5) are gained by applying these three steps. The fourth step is the most obvious: to solve V and $\sigma_v$ in formula (A2). The distance-to-default is shown in the formula below:

$$DD = \frac{ln(V/F) + (\mu - 0.5\sigma_v^2)T}{\sigma_v\sqrt{T}} \tag{A6}$$

where $\mu$ is the estimate for the expected annual return of assets. The implied probability of default is the expected default frequency (EDF), which can be written as:

$$\pi_{Merton} = N\left(-\left(\frac{ln(V/F) + (\mu - 0.5\sigma_v^2)T}{\sigma_v\sqrt{T}}\right)\right) = N(-DD) \tag{A7}$$

If the assumptions of the Merton model are agreed upon, then the Merton DD model can forecast the accurate probability of default. As a matter of fact, if the Merton model's assumptions can be ensured, the implied probability of default should be presented as π_merton, which is a valid statistic for the prediction of default; therefore, this research first examines this assumption.

It is rational and expedient to solve formulas (A2) and (A5) at the same time; however, Crosbie and Bohn (2002) claimed that the effect of market leverage would lead to an irrational result in the solution of formula (A5). Crosbie and Bohn (2002) and Vassalou and Xing (2004) provided a complicated method to solve this problem: First, assume the initial value is $\sigma_v = \sigma_E[E/(E+F)]$, then use $\sigma_v$ to infer the market price of daily company assets in the previous year from formula (A2). After calculating the implied return of daily assets, use this series of returns to find the new estimated value of $\sigma_v$ and $\mu$. Calculate the

$\sigma_v$ repetitively until the convergence is smaller than 0.001. This procedure gains the value of π_merton and uses formula (A7) to calculate the relative implied probability of default.

To examine whether π_merton is beneficial for reduced-form models, this research follows the work of and Bharath and Shumway (2008) and constructs a simplified alternative probability (π_shumway) without solving formulas (A2) and (A5) as the Merton DD model does above. The construction of π_shumway has two purposes: First, it provides the same data as the Merton DD model; it is thus hoped that π_shumway will have the same forecasting ability as π_merton and become the functional form of π_merton. Second, formulas and complicated estimations do not have to be solved if π_shumway can be simplified. The procedure for estimating or optimizing π_shumway is described below.

Before the construction of π_shumway, assume the market price of each company's liability is equal to the face value of its liability:

$$\text{Shumway } D = F \tag{A8}$$

If the liability risk of a company increases, the company is closer to the point of default and bankruptcy. If the risk of liability is related to the risk of shareholder equity, the volatility of each company's liability can be shown as:

$$\text{Shumway } \sigma_D = 0.05 + 0.25 * \sigma_E \tag{A9}$$

As shown in (9), the term structure volatility is set as 5%. In addition, the equity volatility can be set as 25%, so the combined volatility and default risk are related. The total volatility of a company can be shown as:

$$\text{Shumway } \sigma_v = \frac{E}{E+F}\sigma_E + \frac{F}{E+F}(0.05 + 0.25 * \sigma_E) \tag{A10}$$

Afterward, it is assumed that the expected return on company assets will be equal to the stock return over the previous year:

$$\text{Shumway } \mu = r_{it-1} \tag{A11}$$

In order to use the return data of the previous year to simplify the estimate of μ, this research takes the information similar to the Merton DD model and simplifies the distance-to-default (Shumway DD). This can be shown as:

$$\text{Shumway } DD = \frac{ln((E+F)/F) + \left(r_{it-1} - 0.5\text{Shumway}\sigma_v^2\right)T}{\text{Shumway}\sigma_v\sqrt{T}} \tag{A12}$$

The resulting simplified alternative model is easy to calculate, retains the structure of the Merton DD distance-to-default and the expected default frequency. It also employs the same method as the Merton DD model. Using this alternative model can be helpful for solving and calculating π_merton. The further estimation of simplified probability is:

$$\pi\_shumway = N(-\text{shumway DD}) \tag{A13}$$

Commenting on π_shumway is simple because the term structure and equity volatility can be set according to different limitations of data characteristics. The goal is to construct a predictive tool with simple calculations and significant statistical results. If π_shumway has the same predictive result as π_merton, it will be possible to carefully construct a probability that has better predictive ability than π_shumway and covers the same scope of data.

**Note**

[1] Taiwan Economic Journal (TEJ) defines financial distress as including the following nine situations: bounced check, bank run, shutdown and bankruptcy, CPA opinion (for instance, accountants hold negative opinions toward continuous operation, or they

doubt the assumption of continuous operation.), rearrangement, asking for bailout (but if a company only requires an interest decrease, it is not financially distressed), taking over, total delisting (excluding companies whose book value per share is less than 5 dollars), work stoppage because of financial problems, negative net worth, etc. Quasi-Financial distress indicators include appropriating and draining company fund, temporary shutdown, a bounced check by the president of a company, shrinking banks, severe deficit, work stoppage because of economic downturn, and a decrease in value, etc.

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
