# Peer review of "Financial Distress Prediction of Cooperative Financial Institutions—Evidence for Taiwan Credit Unions"

_ijfs, doi:10.3390/ijfs10020030_

Round 1
Reviewer 1 Report
The paper examines the determinants of financial distress for Taiwanese credit unions by considering Merton DD variables beyond accustomed predictors. It first researches the idiosyncrasies of credit unions and provides a literature review of earlier methodological and empirical findings. Then it develops alternative models to predict the failure of Taiwanese credit unions.
Although a great number of intellectual contributions can be detected in the paper, I have to remark that the quality of the paper needs substantial improvement to be considered to be published in IJFS. These are the followings:
- Specify the positioning of the paper in academic literature in the Introduction.
- Data from between 2001 and 2009 are very obsolete. I am skeptical about drawing generalizable conclusions from them. Please explain why you think that this data collection meets the requirements of the journal.
- Please explain on page 2 why “leverage ratios capture operational risk, interest rate risk and reputation risk to be better predictors of failure over short time periods than the more sophisticated risk-based capital ratios, which focus primarily on credit risk”. How do leverage ratios relate to operational and interest rate risk?
- Subsection 2A on the bottom of page 2 finishes with 9 references to literature of bank failure prediction, however, the subsection is named corporate failure. Please reformulate the whole subsection to cover literature review of bank failure prediction, as it is more important for the purposes of this paper than corporate failure.
- Specify on page 4 how to compare the performance of market-based failure prediction models with multivariate statistical classification models.
- Section 3 needs improvement, since it contains key theoretical considerations and methodological findings, however, no reference was made to relevant academic literature.
- Define exactly in Section 4A what financial distress and quasi financial distress means, as it is crucial in creating target variable. It is more important than just placing a footnote without indicating who, when and how defined it in the Taiwan Economic Journal.
- Explain in Section 4A what meant extreme value to be eliminated from the database. Regarding missing value handling, did you discard records having just one missing data, or the ones having completely missing data?
- I have to remark that EDF of Merton and Shumway can be regarded rather as a target variable than explanatory variable when developing failure prediction models. It has led to false regression, unfounded model design, and such trivial findings that “It indicates that the higher is the credit union’s default, the more likely is the credit union to have financial distress”.
- In certain models results indicate that inter alia older, larger, and more profitable credit unions might face higher risk of financial distress, which can be the consequence of the previous remark (unfounded model design), rather than reflecting reality.
- The paper applied the original version of Merton model developed to corporates; however, it has been already customized to banks for a long time, available in academic literature. Please note that credit unions are not companies.
- It has to be analyzed more in-depth what stands behind the declining LOAN RATIO to the extent presented in the paper, what factors led to increasing capital positions (RES CAP doubled in the analyzed period), any why LAB COST ratio increased substantially in Taiwanese credit unions.
- Based on Table 4 the portfolio showed a 16.88% cumulated default rate in 9 years. Please explain why default behavior is not monotonic after 2005, as it constantly decreases from 17.42% which is not intuitive, especially in the era of the global financial crisis. Decreasing pace of growth would be proper instead of declining cumulated default rate showing negative marginal probability of financial distress.
- Parameter signs in Table 6 do not follow professional standards in case of a number of variables. Please review model development, as it is visible that such many variables cannot be entered to arrive at interpretable models. In addition, explain why you entered the standard deviation of other models into the developed models.
- Problem is similar in Table 7 with the testing of the three model designs on the right. Accordingly, results are not convincing.
- Predictive power (=model performance, =classification accuracy) of the models was not tested, only the significance of variables. Do not mix model performance or classification accuracy with results of Hausman test or R-squared statistics.
- I do not believe that it currently has added value to demonstrate what became worse between 2007 and 2009 compared to earlier years. More than 10 years have elapsed since the Great Financial Crisis. It would be interesting, however, to examine more recent data with this regard.
- Define abbreviations when mentioning them for the first time (DD, DEA…).
- Indicate in the body text where you wish to place the tables in Annex (Insert Table 1 here…).
Reviewer 2 Report
Kindly refer to the following comments and suggestions for improvement.
1) The authors should include more latest references to support this study
2) The authors should elaborate more on how to determine the period of study (2001-2009)
3) Check the alignment of the equations
4) Suggested to include the tables in section 4 & 5 and not after references to improve readibility
5) Wrong numbering for conclusions section (should be 6 and not 5)
6) The authors should elaborate more on the significance of this study in the introduction part
Round 2
Reviewer 1 Report
Authors have made efforts to improve the quality of the paper. Hence, I can accept the responses and revisions covering points 1, 2, 4, 5, 6, 8, 11, 12, 13, 14, 16, 17, and 18.
My remaining comments and recommendations are the followings:
- Point 3: It is fully understandable why authors applied leverage ratio in the models. I just remark that leverage ratio has nothing to do with operational, reputational, and interest risks. So please put the ratio into the right context.
- Point 7: Not done. There is no literature reference to the definition of financial distress used by Taiwan Economic Journal.
- Point 9: I am not against entering EDF into the models, however, only up to the point, when the model designs are in compliance with economic interpretability and professional standards.
- Point 10: I understand your logic of entering variables into the models. Which is viable when performing multivariate explorative analysis, and find relationships between variables. However, it is not acceptable when developing prediction models (check Table 7). Predictive models should not contain insignificant variables, and significant variables either that show inappropriate parameter signs (e.g. GDP has positive impact on financial distress in the logit model). Please review the models in line with these requirements and re-evaluate the results.
- Point 15: Not done. Predictive performance indicators of the models were not compared and evaluated in the paper.
